# In Vitro Digestion of Polyphenolic Compounds and the Antioxidant Activity of Acorn Flour and Pasta Enriched with Acorn Flour

**DOI:** 10.3390/ijms25105404

**Published:** 2024-05-15

**Authors:** Kamila Kasprzak-Drozd, Jarosław Mołdoch, Marek Gancarz, Agnieszka Wójtowicz, Iwona Kowalska, Tomasz Oniszczuk, Anna Oniszczuk

**Affiliations:** 1Department of Inorganic Chemistry, Medical University of Lublin, Chodźki 4a, 20-093 Lublin, Poland; kamilakasprzakdrozd@umlub.pl; 2Department of Biochemistry and Crop Quality, Institute of Soil Science and Plant Cultivation, State Research Institute, Czartoryskich 8, 24-100 Puławy, Poland; jmoldoch@iung.pulawy.pl (J.M.); ikowalska@iung.pulawy.pl (I.K.); 3Faculty of Production and Power Engineering, University of Agriculture in Krakow, Balicka 116b, 30-149 Kraków, Poland; m.gancarz@urk.edu.pl; 4Institute of Agrophysics Polish Academy of Sciences, Doświadczalna 4, 20-290 Lublin, Poland; 5Center of Innovation and Research on Healthy and Safe Food, University of Agriculture in Kraków, Balicka 104, 30-149 Kraków, Poland; 6Department of Thermal Technology and Food Process Engineering, University of Life Sciences in Lublin, Głęboka 31, 20-612 Lublin, Poland; agnieszka.wojtowicz@up.lublin.pl (A.W.); tomasz.oniszczuk@up.lublin.pl (T.O.)

**Keywords:** gastrointestinal digestion, gluten-free food, functional food, *Quercus suber* L., flavonoids, phenolic acids, chromatography

## Abstract

Acorn flour is a rich source of nutrients and is beneficial to human health due to, among other things, its low glycemic index and polyphenol content. In order to obtain more accurate data on the levels and activities of the substances tested after ingestion and digestion, it may be beneficial to use a simulated in vitro digestion method. Therefore, the objective of the present study was to elucidate the content of polyphenols, individual phenolic acids, flavonoids and antiradical properties of acorn flour and pasta enriched with acorn flour before and after simulated in vitro gastrointestinal digestion. The results indicate that the total polyphenol content (TPC), flavonoid content and radical scavenging activity exhibited an increasing trend following the initial digestion stage and a decreasing trend following the second stage. Nevertheless, the levels of phenolic acids demonstrated an increase in both digestion phases. The digestion processes of polyphenols in acorn flour differ significantly from those in pasta. In the case of pasta, total polyphenols, phenolic acids and flavonoids, as well as free radical scavenging properties, demonstrated a decreasing trend following each digestion stage.

## 1. Introduction

Polyphenolic compounds are a group of bioactive compounds found in various plant-based foods and beverages. They have been associated with numerous health benefits, including antioxidant and anti-inflammatory properties. Acorn flour (from *Quercus suber* L. acorn) is a rich source of polyphenols. It offers a range of health benefits, including providing a good source of dietary fiber, protein and healthy fats. Additionally, it contains vitamins and minerals, including vitamin E, potassium and magnesium, as well as polyphenols [1]. Furthermore, it is low in carbohydrates and possesses a low glycemic index, rendering it suitable for those with diabetes or those following a low-carbohydrate diet. It can be used as a gluten-free alternative for regular flour in baking, such as the production of bread, pancakes or cookies. It can impart a distinctive nutty flavor and texture to dishes [2]. Acorn flour has been used for centuries in various cuisines around the world, particularly in regions where oak trees are abundant.

Simulated gastrointestinal digestion is a laboratory technique used to mimic the process of digestion that occurs in the human gastrointestinal tract. It involves subjecting food samples to a series of conditions that simulate the different stages of digestion, including the action of enzymes and the acidic environment of the stomach. The purpose of simulated gastrointestinal digestion is to study the breakdown of food components and assess their bioavailability and potential health effects. It allows researchers to understand how different foods are digested and how their nutrients are released and absorbed by the body [3]. During simulated in vitro gastrointestinal digestion, polyphenolic compounds can undergo various changes due to the action of digestive enzymes, pH variations and interactions with other food components. The stability of polyphenols during digestion depends on several factors, including their chemical structure and concentration and the specific conditions of the digestion model. The stability of polyphenols can also be influenced by the presence of other food components. For instance, the presence of proteins and carbohydrates can form complexes with polyphenols, potentially affecting their stability and bioavailability [4,5,6,7,8,9].

Most research is based only on the determination of the concentration of phytonutrients in food products. These studies do not provide complete information on the actual bioavailability of these components [10,11,12,13,14]. 

Thus, the aim of this study was to determine the contents of polyphenolic compounds, flavonoids and individual phenolic acids, as well the antiradical properties, of acorn flour and semolina pasta enriched with acorn flour before and after in vitro simulated gastrointestinal digestion. Our research represents the first comprehensive investigation into the in vitro digestion of acorn flour and pasta enriched with this flour. It is important to note that the results obtained from in vitro digestion models may not fully reflect the actual digestion and absorption of polyphenols in the human body [15,16]. In vivo studies are necessary to provide a more accurate understanding of the fate and bioavailability of polyphenolic compounds.

## 2. Results and Discussion

### 2.1. Nutrient Composition of Acorn Flour

The results of previous studies have proven that acorn flour is rich in nutrients and has many health benefits [17,18]. It is an abundant source of dietary fiber, protein and healthy fats. According to Vinha et al. [17], acorn oil has a promising properties because it presents high similarity with olive oil. Table 1 contains information about nutritional values of acorn flour. 

The results obtained by the authors confirm the results of previous studies. Acorn flour had a significant fiber content (30.07%), which was much higher than in the study by Masmoudi et al. [18]. The fat content was around 9%, which is in line with the literature’s results, where it can range from 2 to 30%. Flour had also moderate contents of proteins and ash (4.33 and 1.90%, respectively), which is consistent with findings presented by Masmoudi et al. [18] (4.20% proteins and 1.62% ash).

This study showed a high content of carbohydrates (45.41%), including starch. This carbohydrate richness has already been reported in the literature for many acorn varieties, with this component’s content being more than 50% [17]. The high starch content of acorn flour may provide it with good functional characteristics, such as viscosity, swelling and gelling properties associated with this compound. According to Correia et al. [20], acorn fruits could be an interesting source of starch for the food industry without the need for chemical or genetic modification. These authors reported that acorn starch has high paste stability during heating and shearing, generating strong and stable gels.

### 2.2. Effect of the In Vitro Digestion on the Contents of Polyphenolic Compounds, Flavonoids, Free Phenolic Acids and Antiradical Properties of Acorn Flour

The studies that solely determine the concentrations of phytonutrients in food products do not provide sufficient information on the actual bioavailable contents of these active ingredients. In vitro digestion experiments offer more detailed information on the levels of test ingredients after ingestion and digestion in the digestive tract [21]. Polyphenols are a group of compounds found in plant-based foods that have been associated with various health benefits. However, the bioavailability and activity of polyphenols can be influenced by the process of digestion. During digestion, polyphenols undergo several changes that can affect their stability and bioavailability. 

The first stage of digestion occurs in the stomach, where hydrochloric acid and pepsin help to denature proteins and break down polyphenols. Some polyphenols may be absorbed in the stomach, while others can be conjugated with glucuronic acid [9]. The next stage of digestion takes place in the small intestine, where the bioavailability of polyphenols can be influenced by various factors. The concentrations of polyphenols, as well as the presence of enzymes in the intestinal matrix, can affect the digestion process. Some polyphenols may undergo degradation or isomerization in the presence of oxygen. The absorption of polyphenols in the small intestine can occur through passive diffusion or active transport [16]. In the large intestine, polyphenols are further degraded by colonic fermentative bacteria. Ultimately, the metabolites of polyphenols lead to the formation of benzoic acid. It is important to note that the bioavailability and bioaccessibility of polyphenols can vary depending on factors such as the food matrix, processing methods and individual differences in digestion and metabolism [22,23]. 

In our study, we used a two-stage in vitro digestion model, including gastric and duodenal phases. The authors were interested from the outset in approximating the bioavailability/release of polyphenols rather than determining the digestibility of nutrients, so we chose a simplified protocol in relation to INFOGEST. We also decided to dispense with the oral phase of digestion because, after initial trials, it became apparent that virtually no changes were taking place in the oral cavity for the polyphenolic compounds that we tested. Due to the usually very short interaction of oral enzymes with the food bolus prior to reaching the stomach, their influence is much less clear and rather limited to carbohydrate-rich products.For example, it is estimated that only 5% of the consumed starch is degraded in the mouth cavity by salivary amylase foods. Therefore, a static in vitro digestion model comprising the two stages of digestion (gastric and duodenal), as proposed by Egger et al. [24], with some modifications, was applied. This type of digestion model has been used by many authors in the context of the study of polyphenolic compounds [1,9,24,25,26,27]. 

Before carrying out in vitro digestion, we examined the total contents of polyphenolics and flavonoids, as well as free phenolic acid content and the radical scavenging activity of the samples. We observed that the total contents of polyphenols slightly increased after the gastric digestion of flour and decreased after duodenal digestion. Similarly, the contents of flavonoids and radical scavenging activity increased after the first stage of digestion and decreased after the second stage (Table 2).

A reduction in TPC following in vitro gastrointestinal digestion has also been reported by other authors who have addressed this issue in lettuce [24] and burdock root flour [25]. Additionally, Sánchez-Gutiérrez et al. [23] observed that following an increase in TPC levels during the gastric phase, there was a significant decrease in polyphenol content during the simulated gastrointestinal digestion of ground *Quercus ilex* leaves. Dacrema et al. [26] reported a loss of polyphenolic compounds in commercial bioactive *Epilobium angustifolium* L. extract after the duodenal phase in the range of 11.83–98.07%. According to these works, phenolics are highly sensitive to pH changes, which affect the stability of bioactive compounds after gastrointestinal digestion. 

Our study found that the antioxidant properties of acorn flour increased after the digestive stage in the stomach but decreased slightly after the second stage in the intestine. This decrease in the antioxidant activity of acorn flour is in line with the results of other studies. Another study investigated the effect of in vitro digestion on the antioxidant capacity of single phenolic compounds. The study found that after the duodenal phase, the antioxidant potential of rutin remained unchanged, whereas the activity of caffeic acid decreased from 12% to 19% and that of rosmarinic acid from 24% to 36% [27]. Sweet potato leaf extracts rich in caffeic and chlorogenic acids [28] showed similar results. The study also found that alkaline pH had a negative effect on antioxidant activity. Neutral and weakly acidic environments led to increased activity. Rodríguez-Roque et al. [29] conducted the in vitro digestion of a soy beverage that is rich in phenolic compounds. They observed a higher antioxidant capacity after gastric digestion compared to intestinal digestion. Likewise, for honey that contains flavonoids (such as luteolin, apigenin, hesperitin and quercetin) and phenolic acids (such as benzoic acid, cinnamic acid, vanillic acid, caffeic acid, coumaric acid, ellagic acid, and chlorogenic acid), a notable decrease in activity was observed by the authors as the pH increased [30]. Structural rearrangements of certain compounds that are sensitive to alkaline pH may be responsible for the reduction in antioxidant capacity under intestinal conditions. In addition, these compounds have the ability to bind to other components of the matrix, resulting in the formation of complexes that may also lead to a reduction in their antioxidant activity. It is possible that the esterification with glucuronic acid and sulphation of phenolic compounds affect their hydrophobic properties and electron delocalization ability, so the antioxidant properties of such modified compounds may be different from those exhibited by analogous aglycones. [9]. Moreover, the correlation between the antioxidant potential of plant material and the number and position of OH groups in the main compounds and their hydrogen-donating capacity has been demonstrated. Therefore, to explain the effect of pH on the antioxidant activity of polyphenol-rich foods, it is necessary to consider each component [31].

As previously stated, the presence of phenolic compounds is linked to antioxidant activity, indicating a significant correlation between polyphenols and antioxidant capacity. Sánchez-Gutiérrez et al. [23] discovered a significant positive correlation between the total polyphenol content and antioxidant properties in the leaf extract of *Q. ilex* (r^2^ = 0.972) and ground leaves (r^2^ = 0.963), as well as in the ORAC test in *Q. ilex* leaf extract (r^2^ = 0.978) and ground leaf (r^2^ = 0.918). These findings suggest that polyphenols play a significant role in the free radical scavenging properties of both matrices. The obtained correlation values are consistent with previous studies that have shown a strong correlation between polyphenolic compounds and antioxidant activity [32,33]. Furthermore, significant correlation was observed between ABTS+ and ORAC tests in both *Q. ilex* leaf extract (r^2^ = 0.939) and ground leaves (r^2^ = 0.919) [23].

The digestion of free phenolic acids showed a different trend. The extracts contained five phenolic acids before digestion: protocatechuic, *p*-OH-benzoic, vanillic, syringic and *p*-coumaric (Table 3). After the first stage of digestion, the same acids were identified. After the second digestion step, caffeic acid was also detected. The levels of all acids increased during the gastric phase of digestion and continued to increase during the intestinal phase.

The results suggest that phenolic compounds are released in the digestive environment of the stomach through the cleavage of bonds with dietary components such as proteins or fiber. The concentration of free phenolic acids after intestinal digestion increases, and caffeic acid appears due to the enzymatic hydrolysis of bonds between acids and proteins or fiber in the primary matrix. This leads to the release of these phenolic compounds [34,35]. This is supported by a study by Sánchez-Gutiérrez et al. [23], who, during the gastric phase of digestion of ground *Q. ilex*, observed an increase in gallic acid, catechin, epicatechin and rutin, while p-coumaric acid decreased compared to the initial phase. At the end of the final phase, only three compounds were detected: an increase in rutin, a decrease in luteolin and no change in epicatechin.

The results obtained by Majdoub et al. [36] were different. During in vitro digestion in the stomach and duodenum, they observed a decrease in the levels of certain compounds. In particular, the level of caffeoylquinic acid was significantly reduced after 20 min of digestion in the stomach and duodenum. Due to degradation in the gastric and duodenal compartments, coumaroylquinic acids remained only slightly present during simulated digestion. Furthermore, quinic acid was observed only in samples obtained during duodenal digestion, suggesting that this compound was formed via the degradation of the more complex caffeoylquinic and coumaroylquinic acids. In the case of rutin, no significant differences were observed between the initial amount and the amounts recovered during the gastric phase. During the gastric phase of the in vitro digestion of black carrots, there was a reduction in phenolic acids such as chlorogenic, neochlorogenic and cryptochlorogenic acids. This reduction intensified in the later stages of digestion, while ferulic and caffeic acids had higher levels compared to undigested samples [37].

In plants, glycosylated, polymerized and esterified forms of phenolic compounds are predominant. They are, therefore, subject to hydrolysis during digestion in the acidic environment of the stomach, the alkaline environment of the intestine and under the influence of digestive enzymes. These conditions cause changes in the structures of phenols, including hydroxylation, glycosylation and dimerization, as well as partial degradation of their primary structures [9]. It is likely that the final step in the transformation of phenolic acids is their combination with glucuronic acid or sulphation, while in the case of flavonols and flavanols, the modification involves methylation. Both glycosides and aglycones are absorbed in the bodies of humans. Due to the many variables that can affect digestion in the gastrointestinal tract, comparison of bioavailability studies is difficult. The influence of the plant and food matrices, the heterogeneity of the plant materials analyzed and the in vitro digestion methodology can lead to differences in results. For instance, rutin can experience a loss of anywhere between 3% and total loss during intestinal digestion, while quercetin can experience a loss ranging from 5.8% [38] to total loss [22]. Similarly, chlorogenic acid can experience a loss ranging from 44% to 95.7% [22]. Pure compounds exhibit significant variability in terms of their bioavailability. It is widely acknowledged that the methodology used for digestion plays a crucial role in determining the bioavailability of polyphenols.

### 2.3. Effect of the In Vitro Digestion on the Contents of Polyphenolic Compounds, Flavonoids, Free Phenolic Acids and Antiradical Properties of Pasta Enriched Acorn Flour

In the subsequent phase of the experiment, the authors developed a semolina pasta enriched with acorn flour and tested it before and after in vitro digestion for polyphenolic compounds, flavonoids, free phenolic acids and antiradical properties, taking into account the needs and expectations of consumers. Semolina is a popular ingredient in the kitchen because it has several health-promoting properties. It is a good source of carbohydrates and protein and contains a significant amount of fiber, which helps digestion and makes you feel full. Semolina is rich in minerals such as phosphorus, magnesium, iron and copper. It contains B vitamins including folic acid, vitamin B1 and niacin. Semolina products have a lower glycemic index than those made with wheat flour. This means that they raise blood sugar levels more slowly and gradually. For people with diabetes and obesity, but also for healthy people who want to maintain a healthy body weight, foods with a high glycemic index are important [9].

It was found that even a small addition of acorn flour increased pasta’s polyphenol content (including flavonoids and free phenolic acids) and antioxidant activity (Table 4 and Table 5).

It can easily be seen that the digestion of polyphenolic compounds in acorn flour is quite different from that in pasta, both with and without acorn flour. In the pasta, the total content of polyphenols, the content of flavonoids and (in particular) the free radical scavenging properties are drastically reduced after each digestion step (Table 4). In both types of pasta, before digestion, the presence of three phenolic acids was detected: protocatechuic, *p*-OH-benzoic and vanillic acids (Table 5). Two of these acids, *p*-OH-benzoic acid and protocatechuic acid, were no longer observed after the digestion, whereas the content of vanillic acid decreased successively after the two digestion steps. The different structure of the matrices is responsible for the differences in the digestion of the polyphenolic compounds present in acorn flour and those present in pasta. Studies have shown that the phenolic compounds can bind to the starch during the digestion of pasta [39]. Furthermore, tannins (together or with other food components, particularly carbohydrates and proteins) form polymers that are more difficult to extract [40]. The interactions of tannins with food components are mostly non-covalent and may involve hydrogen bonding and hydrophobic interactions. Tannins found in acorn flour have a strong affinity for prolamin-containing proteins [41]. During the digestion process, condensed tannins are thought to be damaged or structurally altered. Tannins exposed to low pH and elevated temperature depolymerize as oligomers and monomers and the basic phenolic structure remains stable, according to Dlamini [42]. Condensed tannins polymerase and higher molecular weight cross-linked polymers are formed when exposed to a higher pH. Performing a principal components analysis (PCA) allowed us to obtain six new variables’ principal components, which explain 100% of the variability in the system. Figure 1a shows the projection of the variables on planes PC1 (84.04%) and PC2 (11.51%), which describe the dependencies at 95.55%. A strong positive correlation was found between *p*-OH-benzoic acid and protocatechuic acid. The correlation between flavonoid content, radical scavenging activity and polyphenol content is also strong and positive.

The correlation between *p*-OH-benzoic acid, protocatechuic acid, flavonoid content, radical scavenging activity, polyphenol content and vanillic acid is positive but weak (Figure 1a,b, Table 6). All compounds strongly affect the determination ability of the pasta before and after two-stage digestion and with or without the acorn flour. Table 6 showed the correlation matrix of the effects of total content of polyphenolic compounds and flavonoids, as well as radical scavenging activity. PCA analysis was also used to describe the effect of the total contents of polyphenolic compounds and flavonoids, as well as radical scavenging activity. The determinant of the correlation matrix estimates the collinearity (correlation) of the explanatory variables. Closer to 0, the degree of mutual correlation of the explanatory variables was lower, whereas being closer to 1 made the stronger correlation. The sign determines whether the direction of the correlation is positive or negative.

Figure 1b shows cases of pasta before and after two-stage digestion and with or without the acorn flour. Positive values of PC1 describes cases of pasta after two-stage digestion and with or without the acorn flour. Negative PC1 values describe parameters before digestion with and without acorn flour. Positive values of PC1 with PC2 describe cases of pasta after two-stage digestion and without the acorn flour. Negative PC2 values describe cases of pasta after two-stage digestion but with the acorn flour.

Differences in the digestion of polyphenolic compounds during the digestion of acorn flour and pasta are clearly visible in the correlations presented for individual groups of compounds (Table 7). However, despite the differences, for all samples, one regularity can be seen: their antioxidant properties are in high positive correlation with the levels of flavonoids.

Flavonoids are known to prevent free radical damage in several ways, one of which is the direct scavenging of free radicals. This process involves oxidizing the free radicals to more stable, less reactive radicals, thereby stabilizing the reactive oxygen species. The high reactivity of the hydroxyl group of flavonoids is thought to inactivate the radicals by reacting with them [43]. Some flavonoids are able to directly scavenge superoxides, while others are able to scavenge a highly reactive oxygen radical, called peroxynitrite. This was found by Hanasaki et al. [44]. It was also observed that flavonoids such as epicatechin and rutin are potent radical scavengers and that the scavenging ability of rutin may be due to its inhibitory activity on the enzyme xanthine oxidase. The authors also mentioned that this action protects LDL (low-density lipoprotein) particles, and, theoretically, flavonoids may have a preventive effect against atherosclerosis. Moreover, the findings demonstrated that silibin and quercetin inhibit xanthine oxidase activity, resulting in reduced oxidative stress [43]. This is one of the many reasons why the incorporation of high concentrations of naturally occurring polyphenols, in particular flavonoids, into food products is of significant importance.

## 3. Materials and Methods

### 3.1. Chemicals

Formic acid, acetonitrile and ethanol were purchased from J.T. Baker (Phillipsburg, NJ, USA) for chromatography and extraction purposes. The Folin–Ciocalteu reagent was also obtained from J.T. Baker. Sigma Aldrich (St. Louis, MO, USA) provided the standards for 2,2-diphenyl-1-picrylhydrazyl (DPPH), hydrochloric acid, sodium bicarbonate, sodium glycodeoxycholate, sodium taurocholate, phenol, sodium taurodeoxycholate, pancreatin from porcine pancreas (code P7545; 207 protease units mg^−f^ solid; 238 α-amylase units mg^−s^ solid; and 29.9 lipase units mg^−g^ solid) and pepsin from porcine gastric mucosa (code P7000; 474 units mg^−s^ solid).

### 3.2. Pasta Production

The semolina and the acorn flour were bought in a health food shop. The pasta without acorn flour was made with semolina, water and salt only, while the pasta with acorn flour was made with semolina (96%), water, salt and acorn flour (4%). Both pasta samples were handmade. The pasta samples are shown in Figure 2. 

### 3.3. Extraction Procedure

Non-digested acorn flour and pasta samples (2 g portions) were extracted using ultrasound assistance in an ultrasonic bath (Bandelin Electronic GmbH & Co. KG, Berlin, Germany) with 80% ethanol (60 mL) at 60 °C for 40 min (ultrasound frequency of 33 kHz and a power of 320 W). The resulting essences were filtered, dried via evaporation and dissolved in methanol (10 mL) [9].

### 3.4. Determination of Phenolic Acids 

The hydrolysis of the samples was carried out according to the modified method of Czaban et al. [15]. Free phenolic acid content was analyzed via reversed-phase ultra-high pressure liquid chromatography on a Waters ACQUITY UPLC^®^ Systems chromatograph (Waters Corporation, Milford, MA, USA) equipped with a photodiode array detector coupled to a triple quadrupole mass spectrometer (Waters ACQUITY^®^ TQD, Micromass, Manchester, UK, GB). Samples were separated on a Waters ACQUITY UPLC^®^ HSS C18 column (1.0 × 100 mm; 1.8 μm) at 30 °C. The mobile phase consisted of 0.1% formic acid in MilliQ water (*v*/*v*) and 0.1% formic acid in acetonitrile (*v*/*v*). The analytes were eluted using a combination of isocratic and gradient steps. The elution (0.50 mL/min) was carried out with the following gradients of solvent B: 0.00–0.50 min, 8% B; 0.50–8.00 min, 8–20% B; 8.00–8.10 min, 20–95% B; 8.10–10.00 min, 95% B; 10.00–10.10 min, 95–8% B; 10.10–12.00 min, 8% B. The sample injection volume was 2.5 μL (full loop mode) [3].

The detection of phenolic acids was performed in negative ionization mode, using a selected reaction monitoring method. The source temperature was 110 °C, while the desolvation temperature was 350 °C. Nitrogen was used as a desolvation gas (a flow of 1000 L/h) and as a cone gas (100 L/h). Argon was used as a collision gas (0.1 mL/min). Collision energies were optimized for particular phenolic acids [45]. Concentrations of phenolic acids were calculated on the basis of the calibration curves in Table 8.

### 3.5. Determination of the Total Contents of Polyphenolic Compounds (TPC)

The TPC was determined according to the modified Folin–Ciocalteu (FC) method [3,9], expressed as mg of gallic acid equivalents (GAE) per g of dry weight. Next, 100 μL of the tested extracts (obtained via extraction—before digestion and after digestion) were placed in 5 mL volumetric flasks. Then, 900 μL of distilled water and 100 μL of the Folin–Ciocalteu reagent were added. The solutions were mixed and set aside. After 4 min, 1 mL of 7.7% sodium bicarbonate and 400 μL of distilled water were added. The contents were mixed and placed in a 40 °C water bath for 50 min. The absorbance of the solutions was then measured using a spectrometer (Genesys 150 UV-VIS Thermo Scientific, Waltham, MA, USA) at a wavelength of 765 nm.

Then, 100 μL of each of the calibration solutions were collected, and 900 μL of distilled water and 100 μL of Folin–Ciocalteu reagent were added and processed as used for the extracts above. A solution without gallic acid was used for a blank experiment. Based on the results obtained, a calibration curve of gallic acid (y = 3.2941x + 0.0314, R^2^ = 0.9992) was plotted. This was used to calculate the total polyphenol content.

### 3.6. Ability to Scavenge DPPH

The measurement of antiradical activity of samples before and after digestion was carried out via DPPH stable radical (2,2-diphenyl-1- picrylhydrazyl) spectroscopy [3].

Absorbance was measured at a wavelength of 517 nm, and the UV-VIS spectrophotometer (Genesys 150 UV-VIS, Thermo Scientific, Waltham, MA, USA) was calibrated to pure methanol. Measurements were performed every 5 min until a plateau was reached. This allowed the changes in absorbance to be monitored over time and indicated when the plateau was reached. The following formula was used to calculate the radical scavenging potential of the extracts:$$\%RSA=\left[\frac{(A_{0}-A_{1)}}{A_{0}}\right]\times 100$$
*A*_0_—the absorbance of the sample except for the tested extracts;*A*_1_—the absorbance of the sample with tested extracts.


### 3.7. Moisture, Protein, Ash, Carbohydrates, Lipid and Fiber Contents

The determination of moisture content was carried out according to the modified method of Pontieri et al. [4]. A ceramic capsule was accurately weighed after complete desiccation at 100 °C in a vacuum using an oven (ISCO mod. NSV9035, Milan, Italy) and left at room temperature in a silica gel dryer. A sample of the acorn flour was placed in a container of ceramic material. The sample was dehumidified by keeping it at the same temperature and pressure until a constant weight was reached. The measurement of the moisture content was the loss of weight of the sample. The Kjeldahl method [5] was used to determine the nitrogen content. Flour samples were analyzed using a Mineral Six digester and an Auto Disteam semi-automatic distiller (International PBI, Milan, Italy).

To measure total ash, flour samples were incinerated at ~550 °C, and the dishes placed in a desiccator to cool. After being brought to room temperature, the dishes were weighed [6]. Carbohydrate content was measured via subtraction as the amount of material remaining after accounting for fat, ash, moisture and protein contents [7]. The lipid content was determined as described by Pontieri et al. [4], albeit with modifications. The samples were crushed with liquid nitrogen, freeze-dried in the FTS Flex-DryTM unit and extracted with chloroform for 4 h in the Soxhlet unit. To determine the amount of fat extracted, the extracts were dried to dryness and weighed. The AOAC method was used to determine fiber content [8].

### 3.8. In Vitro Two-Stage Digestion Model 

Seraglio et al.’s [16] static in vitro two-stage (gastric and duodenal) digestion model was used with modifications. For the development of the gastric digestion phase, 1.632 g of each sample was homogenized, mixed with 5.84 mL of gastric solution and stirred for 4 min. Next, 2.32 mL hydrochloric acid, at pH 2.5 ± 0.2, was added to this mixture. Samples were incubated in a water bath for two hours (37 °C, 100 rpm). The samples were then centrifuged (10 min, 8000 rpm), and the supernatant was collected for further analysis. Samples were refrigerated (−20 °C for 24 h) until the scheduled analysis.

The evaluation of duodenal digestion was the same as for the evaluation of gastric digestion. After incubation, 0.09 mL 1 mol/L sodium bicarbonate (to raise pH to 5.5) and 2.26 mL duodenal solution were added to each sample and stirred for 1 min. After this time, 0.72 mL of sodium bicarbonate solution was added to each flask to adjust the pH to 6.7 ± 0.2. This was followed by incubation in a water bath for 2 h at 37 °C and 100 rpm. After centrifugation (10 min, 8000 rpm), the supernatant was analyzed. The supernatant was stored as described for gastric digestion.

The preparation of the simulated gastric juice was as follows: 0.16 g of pepsin (from porcine gastric mucosa; 474 units mg^−1^ solid) was dissolved in 0.35 mL of 12 M hydrochloric acid and made up to 50 mL with ultrapure water. Duodenum solution was prepared by dissolving 0.25 g pancreatin (207 protease units mg^−1^ solid; 238 α-amylase units mg^−1^ solid; and 29.9 lipase units mg^−1^ solid) with 0.047 g of sodium glycodeoxycholate, 0.0505 g of sodium taurocholate and 0.029g of sodium taurodeoxycholate in 0.25mL of 0.5M sodium bicarbonate in 25 mL of ultrapure water.

### 3.9. Statistical Analysis

All the measurements were conducted with at least in three replications; the results were mean values of these repetitions and standard deviations (SD). Pearson’s correlation coefficient and their significance were evaluated at 0.05 for the tested characteristics.

Statistical analysis with ANOVA (Statistica 13.0, StatSoft Inc., Tulsa, OK, USA) was applied to determine the significance of differences at α = 0.05; multi-factor analysis of variance and the Tukey test were also carried out. Principal component analysis (PCA), analysis of variance and the determination of correlations were performed at a significance level of α = 0.05. The principal component analysis was employed to determine the relationships between the total contents of polyphenolic compounds, flavonoids and free phenolic acids in pasta, before and after two-stage digestion and with or without the acorn flour addition. A heat map of the correlation matrix of the analyzed parameters using the PCA method is also presented as a table that shows the correlation coefficient for all possible combinations of pairs of variables.

The PCA data matrix for the statistical analysis of the results had 6 columns (names of the compounds) and 6 rows (type of pasta before and after two-stage digestion and with or without acorn flour). The input matrix was scaled automatically. The optimal number of principal components obtained in the analysis was determined based on the Cattel criterion.

## 4. Conclusions

This research offers valuable insights into the nutritional composition of acorn flour and its distinctive health benefits. Experimental findings indicate that this flour is a significant source of polyphenols (9.226 mg GAE/g d.w.) and flavonoids (4.987 mg GAE/g d.w.). Moreover, the results demonstrate the high radical scavenging activity of the extracts against DPPH (90.23%). The subsequent phenolic acids were identified in acorn flour: protocatechuic, *p*-OH-benzoic, vanillic, syringic and *p*-coumaric. After the initial digestion step, the same acids were identified. Furthermore, caffeic acid was identified after the second digestion step. These results indicate that there is a moderate increase in the total polyphenol content after flour digestion in the stomach, followed by a decrease after digestion in the duodenum. In addition, the flavonoid content and radical scavenging activity exhibited a gradual rise after the initial phase of digestion, followed by a decline after the second stage. The outcomes demonstrated that a minimal incorporation of acorn flour could elevate the polyphenol content (including flavonoids and free phenolic acids) of the pasta and enhance its antioxidant efficacy. The digestive processes of polyphenolic compounds in acorn flour differ from those observed in pasta, both with and without flour. It has been demonstrated that the polyphenols, phenolic acids and flavonoid contents of pasta, as well as the free radical scavenging properties, are significantly reduced following each digestion step. 

Our research represents the first comprehensive investigation into the in vitro digestion of acorn flour and pasta enriched with this flour. It is well known that plant products are composed of a diverse range of chemical compounds, which are often consumed in conjunction with other food items. Our findings indicate that the composition of a food matrix, as well as its subsequent breakdown during digestion, can significantly influence the bioavailability and stability of phytochemicals. Given that the potential efficacy of plant metabolites for human health is contingent upon their bioavailability, it is of significant importance to maintain a continuous update of the state of knowledge regarding the breakdown of food components during digestion.

## Figures and Tables

**Figure 1 ijms-25-05404-f001:**
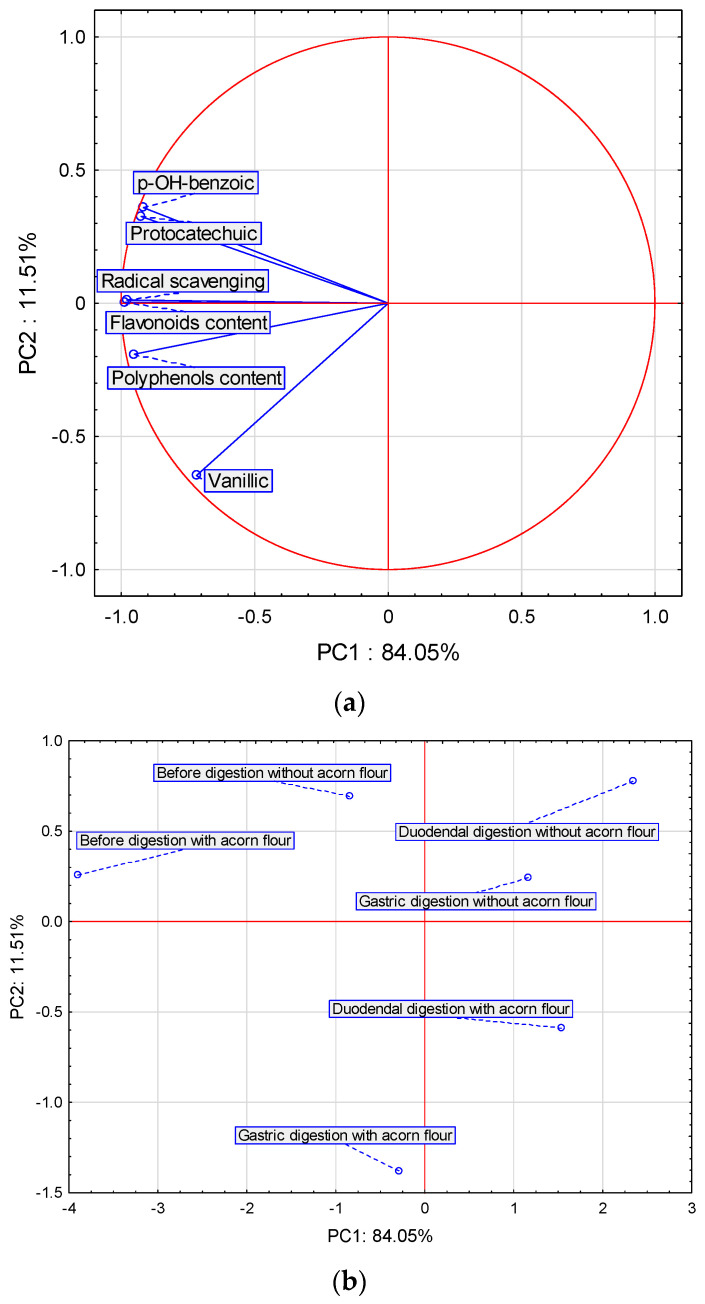
Projection of variables: total contents of polyphenolic compounds and flavonoid contents of free phenolic acids in pasta on the PC1 and PC2 scores plot—(**a**); projection of cases characterizing the pasta before and after two-stage digestion and with the acorn flour on the PC1 and PC2 loadings plot—(**b**).

**Figure 2 ijms-25-05404-f002:**
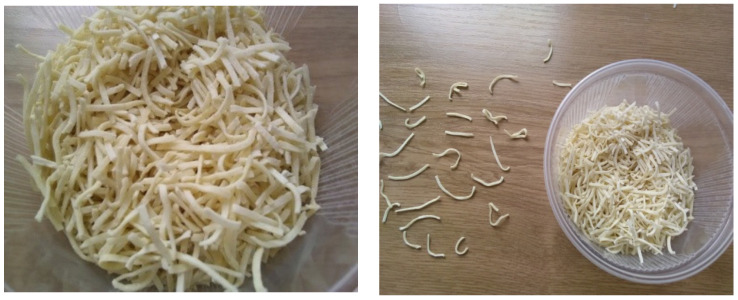
Pasta samples with the addition of acorn flour.

**Table 1 ijms-25-05404-t001:** Nutritional values of acorn flour (number of repetitions *n* = 3; mean ± RSD%).

Parameter	Results (%)	Recommended Daily Dose (g/Day) [19]
Moisture	9.60 ± 0.52	-
Ash	1.90 ± 1.05	-
Total proteins	4.33 ± 4.85	50
Fats	8.69 ± 1.84	70
Total carbohydrates	45.41 ± 2.27	260
Fibers	30.07 ± 2.43	-

**Table 2 ijms-25-05404-t002:** Total contents of polyphenolic compounds and flavonoids (mg gallic acid equivalent (GAE)/g d.w. of extract), as well as the radical scavenging activity [%] of acorn flour before and after two-stage digestion (number of repetitions *n* = 3; mean ± SD).

Parameters	Before Digestion	Gastric Digestion	Duodenal Digestion
Polyphenol content	9.226 ^a^ ± 0.071	9.347 ^a^ ± 0.085	7.176 ^b^ ± 0.112
Flavonoid content	4.987 ^a^ ± 0.003	5.105 ^b^ ± 0.001	4.012 ^c^ ± 0.000
Radical scavenging activity	90.230 ^a^ ± 2.310	91.532 ^a^ ± 0.828	64.748 ^b^ ± 1.982

Identical letters indicate no significant differences between the results obtained after the Tukey test at the significance level α = 0.05. Different letters (a–c) indicate significant differences between the results obtained for the compared compounds and cases.

**Table 3 ijms-25-05404-t003:** The contents of free phenolic acids (µg/g d.w.) in acorn flour before and after two-stage digestion (number of repetitions *n* = 3; mean ± SD).

Phenolic Acid	Before Digestion	Gastric Digestion	Duodenal Digestion
Protocatechuic	3.448 ^a^ ± 0.008	6.528 ^b^ ± 0.011	7.182 ^c^ ± 0.043
*p*-OH-benzoic	1.055 ^a^ ± 0.041	1.738 ^b^ ± 0.008	2.348 ^c^ ± 0.015
Vanillic	0.641 ^a^ ± 0.007	1.790 ^b^ ± 0.025	2.324 ^c^ ± 0.031
Caffeic	-	-	0.062 ± 0.002
Syringic	0.492 ^a^ ± 0.000	1.508 ^b^ ± 0.014	1.815 ^c^ ± 0.058
*p*-coumaric	0.858 ^a^ ± 0.025	1.380 ^b^ ± 0.064	1.391 ^c^ ± 0.055
Sum	6.494	12.944	15.122

Identical letters indicate no significant differences between the results obtained after the Tukey test at the significance level α = 0.05. Different letters (a–c) indicate significant differences between the results obtained for the compared compounds and cases.

**Table 4 ijms-25-05404-t004:** Total contents of polyphenolic compounds and flavonoids (mg GAE/g d.w.), as well as radical scavenging activity [%] of pasta, before and after two-stage digestion (number of repetitions *n* = 3; mean ± SD).

	Parameters	Before Digestion	Gastric Digestion	Duodenal Digestion
Pasta without acorn flour	Polyphenol content	1.258 ^a^ ± 0.004	1.143 ^b^ ± 0.002	0.947 ^c^ ± 0.001
	Flavonoid content	0.364 ^a^ ± 0.001	0.211 ^b^ ± 0.002	0.125 ^c^ ± 0.002
	Radical scavenging	33.244 ^a^ ± 0.151	24.568 ^b^ ± 0.290	5.711 ^c^ ± 0.052
Pasta enriched with acorn flour	Polyphenol content	1.546 ^a^ ± 0.019	1.358 ^b^ ± 0.013	0.998 ^c^ ± 0.008
	Flavonoid content	0.489 ^a^ ± 0.001	0.311 ^b^ ± 0.002	0.161 ^c^ ± 0.003
	Radical scavenging	63.185 ^a^ ± 0.932	31.722 ^b^ ± 0.270	8.295 ^c^ ± 0.161

Identical letters indicate no significant differences between the results obtained after the Tukey test at the significance level α = 0.05. Different letters (a–c) indicate significant differences between the results obtained for the compared compounds and cases.

**Table 5 ijms-25-05404-t005:** The contents of free phenolic acids (µg/g d.w.) in pasta before and after two-stage digestion (number of repetitions *n* = 3; mean ± SD).

	Phenolic Acid	Before Digestion	Gastric Digestion	Duodenal Digestion
Pasta without acorn flour	Protocatechuic	0.061 ± 0.001	-	-
	*p*-OH-benzoic	0.032 ± 0.002	-	-
	Vanillic	0.109 ^a^ ± 0.004	0.083 ^b^ ± 0.004	0.061 ^c^ ± 0.002
	Sum	0.202	0.083	0.061
Pasta enriched with acorn flour	Protocatechuic	0.141 ± 0.002	-	-
	*p*-OH-benzoic	0.059 ± 0.001	-	-
	Vanillic	0.183 ^a^ ± 0.000	0.172 ^b^ ± 0.002	0.144 ^c^ ± 0.001
	Sum	0.383	0.172	0.144

Identical letters indicate no significant differences between the results obtained after the Tukey test at the significance level α = 0.05. Different letters (a–c) indicate significant differences between the results obtained for the compared compounds and cases.

**Table 6 ijms-25-05404-t006:** Correlation matrix for the tested parameters of the effect of acorn flour content at α < 0.05.

	Polyphenol Content	Flavonoid Content	Radical Scavenging	Protocatechuic	p-OH-Benzoic	Vanillic
Polyphenol content	1	0.961	0.971	0.773	0.761	0.744
Flavonoid content	0.961	1	0.972	0.884	0.890	0.675
Radical scavenging	0.971	0.972	1	0.881	0.868	0.650
Protocatechuic	0.773	0.884	0.881	1	0.995	0.501
*p*-OH-benzoic	0.761	0.890	0.868	0.995	1	0.469
Vanillic	0.744	0.675	0.650	0.501	0.469	1
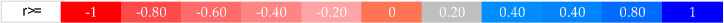

**Table 7 ijms-25-05404-t007:** Pearson’s correlation coefficient at α < 0.05.

Sample	Compound	Total Polyphenols	Free Phenolic Acid	Radical Scavenging Capacity
Acorn flour	Total flavonoids	0.999	−0.619	0.998
	Total polyphenols		−0.659	0.999
	Free phenolic acids			−0.663
Pasta without acorn flour	Total flavonoids	0.952	0.976	0.932
	Total polyphenols		0.864	0.998
	Free phenolic acids			0.832
Pasta enriched with acorn flour	Total flavonoids	0.974	0.933	0.997
	Total polyphenols		0.828	0.988
	Free phenolic acids			0.904

**Table 8 ijms-25-05404-t008:** Parameters of the calibration curve for phenolic acids.

Phenolic Acid	Calibration Curve	R^2^	LOQ (ng/mL)	LOD (ng/mL)
Protocatechuic	y = −0.0254426x^2^ + 1.46612x + 0.0137605	0.997	2.1	0.7
*p*-OH-benzoic	y = −0.0116753x^2^ + 1.43904x + 0.164956	0.997	2.3	0.8
Vanillic	y = 0.000116384x^2^ + 0.194029x − 0.00311	0.998	1.9	0.6
Caffeic	y = −0.0182712x^2^ + 2.42109x + 0.436786	0.995	1.6	0.5
Syringic	y = −0.0000546324x^2^ + 0.259824x − 0.0026428	0.986	1.8	0.6
*p*-coumaric	y = −0.0165714x^2^ + 2.05818x + 2.05818	0.993	0.9	0.3

## Data Availability

The samples and research data are available from the authors.

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
