# Peer review of "In Vitro Digestion of Polyphenolic Compounds and the Antioxidant Activity of Acorn Flour and Pasta Enriched with Acorn Flour"

_ijms, 2024, doi:10.3390/ijms25105404_

Round 1

Reviewer 1 Report

Comments and Suggestions for Authors

Drozd et al reported in the manuscript the preparation and the chemical analysis of Acorn flour and pasta enriched with Acorn flour, with a great attention for polyphenols content. Although this manuscript provides novel insights into enriched pasta preparation and analysis for nutraceutical purposes, the paper needs to be deeply revised. In fact, although the chemical analysis performed and the statistical part of the data is appreciated, the gastro-intestinal digestion protocol used does not agree with what the current harmonized literature (e.g., INFOGEST protocol). In addition, some experimental issues, as the lack of the oral phase digestion, may be reflected in significant effects in the chemical composition of the samples, making unreliable the entire experimental result and all the considerations that the authors reported in the manuscript. Therefore, the authors should repeat the experiments in accordance with the harmonized literature on gastro-simulated digestion protocols. Based on these considerations, although the topic is interesting and could open new development for acorn flour use for pasta preparation, I recommend the rejection of the paper. However, I invite the authors to continue their research to improve it for possible resubmission. Following I reported my revisions:

-        General comment: The main topic of the manuscript is related to the application of an in vitro gastrointestinal digestion protocol. It should be better to add some references to the harmonized INFOGEST protocol, which represents the standardized approach for the study of gastro-intestinal digestion in foods.

-        Material and methods, paragraph 3.1. Chemicals: It should be better to add some details about the enzymatic activity (U/mg) of the enzymes included in the in vitro two-stages digestion protocol.

-        Material and methods, paragraph 3.3. Extraction Procedure: It should be better to clarify the amount (in grams) of acorn flour and pasta samples extracted and the volumes (in ml) of 80% ethanol solution.

-        Material and methods, paragraph 3.8. In vitro Two-Stages Digestion Model: The authors should to clarify the chemical composition of “gastric solution” (line 389) and “duodenal solution (line 396). The two solutions contain only the reagents reported in lines 401-405? Moreover, the classical INFOGEST protocol for gastro-intestinal digestion include the addition of several buffers, including calcium ions which are necessary for enzymes activity.

-        Material and methods, paragraph 3.8. In vitro Two-Stages Digestion Model, lines 392-393: The samples were directly analyzed as digestion supernatants without any concentration/desalting approach? It should be clarified. Moreover, in the paragraph 3.3. Extraction Procedure the authors reported that the samples were extracted with 80% ethanol solution, which suggests that the authors freezed-dried the samples before the extraction.

-        Material and methods, paragraph 3.8. In vitro Two-Stages Digestion Model: The authors reported an in vitro-two stages protocol, which include gastric and duodenal stage. Moreover, they didn’t perform any investigation about the oral phase, which include the use of “salivary amylase” needed only to digest starch-containing foods. The authors reported a great starch content in the samples (45%, line 87) and the possible linkage of polyphenols to the starch during the digestion (line 293). This experimental approach represents a strong limitation of the entire study. The lack of a preliminary step in the digestion step with a specific action on starch represents a problem in the considering of the effects that this could have in subsequent steps (gastrical and duodenal phases) on the real release of phenolic compounds from the samples. It is therefore strongly recommended repeating of digestion experiments including the oral phase and comparing the results with those reported in the current version of the manuscript.

Comments on the Quality of English Language

A moderate editing of English language is required.

Author Response

The authors would like to thank the Reviewer for valuable comments which have helped to improve the quality of the manuscript. We hope that the revisions in the manuscript and accompanying responses will be sufficient to make our manuscript suitable for publication. We have made all the changes suggested in the Reviewer's comments in the text.

Drozd et al reported in the manuscript the preparation and the chemical analysis of Acorn flour and pasta enriched with Acorn flour, with a great attention for polyphenols content. Although this manuscript provides novel insights into enriched pasta preparation and analysis for nutraceutical purposes, the paper needs to be deeply revised. In fact, although the chemical analysis performed and the statistical part of the data is appreciated, the gastro-intestinal digestion protocol used does not agree with what the current harmonized literature (e.g., INFOGEST protocol). In addition, some experimental issues, as the lack of the oral phase digestion, may be reflected in significant effects in the chemical composition of the samples, making unreliable the entire experimental result and all the considerations that the authors reported in the manuscript. Therefore, the authors should repeat the experiments in accordance with the harmonized literature on gastro-simulated digestion protocols. Based on these considerations, although the topic is interesting and could open new development for acorn flour use for pasta preparation, I recommend the rejection of the paper. However, I invite the authors to continue their research to improve it for possible resubmission. Following I reported my revisions:

-        General comment: The main topic of the manuscript is related to the application of an in vitro gastrointestinal digestion protocol. It should be better to add some references to the harmonized INFOGEST protocol, which represents the standardized approach for the study of gastro-intestinal digestion in foods.

The authors are very grateful for your valuable guidance, which we will certainly follow in continuing our research. At the same time, we would like to emphasise that, when embarking on our research, we took great care in reviewing the literature on in vitro digestion methods, both static and dynamic. At the time, we wrote a review paper (Nutrients 2020, 12, 1401; doi:10.3390/nu12051401) entitled 'Influence of In Vitro Digestion on Composition,Bioaccessibility and Antioxidant Activity of Food Polyphenols-A Non-Systematic Review', and then conducted preliminary trials involving different protocols and 3 phases of digestion - oral, gastric and duodenal.

The authors recognise that  INFOGEST is a static, in vitro protocols simulating human digestion that uses constant ratios of meal to digestive fluids and a constant pH for each step of digestion.. Using this method, food samples are subjected to sequential oral, gastric and intestinal digestion while parameters such as electrolytes, enzymes, bile, dilution, pH and time of digestion are based on available physiological data. INFOGEST protocol should be used to assess the endpoints resulting from digestion of foods by analyzing the digestion products (e.g., peptides/amino acids, fatty acids, simple sugars). As we were interested from the outset in approximating the bioavailability/released of active compounds, i.e. polyphenols (the degree to which a compounds can be released and absorbed from the gastrointestinal tract ), rather than determining the digestibility of nutrients (the degree to which a nutrient can be broken down into its constituent parts), we chose a simplified protocol in relation to INFOGEST.

We also decided to dispense with the oral phase of digestion because, after initial trials, it became apparent that virtually no changes was taking place in the oral cavity for the polyphenolic compounds we tested. Due to the usually very short interaction of oral enzymes with the food bolus prior to reaching the stomach, their influence is much less clear and rather limited to carbohydrate-rich. For example, it is estimated that only 5% of the consumed starch is  degraded in the mouth cavity by salivary amylase foods  (Hur and others 2011). Therefore A static in vitro digestion model comprising two-stages of digestion (gastric and duodenal) as proposed by the United States Pharmacopeia (2000), and Egger et al. (2016), with some modifications  was applied.

The article presented here is the third in a series in which the authors have applied a similar methodology to a different range of novel functional food products (https://doi.org/10.3390/molecules28041706 , https://doi.org/10.3390/ijms232214458).

We would also like to point out that this type of digestion model has been used by many authors in the context of the study of polyphenolic compounds, e.g.  doi:10.1111/ijfs.14217, http://dx.doi.org/10.1016/j.foodres.2017.06.024, https://doi.org/10.1016/j.foodchem.2018.02.128,  DOI: 10.15193/zntj/2016/105/121, doi: 10.1111/1541-4337.12081, DOI 10.1002/jsfa.9609, DOI: 10.1021/acs.jafc.9b0224

-        Material and methods, paragraph 3.1. Chemicals: It should be better to add some details about the enzymatic activity (U/mg) of the enzymes included in the in vitro two-stages digestion protocol.

Thank you for your suggestion. The section 3.1 has been updated to include details of the enzymatic activity of the enzymes.

-        Material and methods, paragraph 3.3. Extraction Procedure: It should be better to clarify the amount (in grams) of acorn flour and pasta samples extracted and the volumes (in ml) of 80% ethanol solution.

Thank you for your suggestion. Paragraph 3.3. has been supplemented.

-        Material and methods, paragraph 3.8. In vitro Two-Stages Digestion Model: The authors should to clarify the chemical composition of “gastric solution” (line 389) and “duodenal solution (line 396). The two solutions contain only the reagents reported in lines 401-405? Moreover, the classical INFOGEST protocol for gastro-intestinal digestion include the addition of several buffers, including calcium ions which are necessary for enzymes activity.

The section 3.8. has been revised to include details of the enzymes. Due to the use of a simplified digestion protocol and the focus on polyphenol bioavailability rather than nutrient digestion, we did not add calcium ions (which are activators of lipase and rennet) during the digestion proces. I would like to point out that pH was controlled to be appropriate as described in the cited article (Seraglio et al.) - 2.2 ±  0.2 in gastic digestion and 6.7 ± 0.2 in duodenal digestion.

-        Material and methods, paragraph 3.8. In vitro Two-Stages Digestion Model, lines 392-393: The samples were directly analyzed as digestion supernatants without any concentration/desalting approach? It should be clarified. Moreover, in the paragraph 3.3. Extraction Procedure the authors reported that the samples were extracted with 80% ethanol solution, which suggests that the authors freezed-dried the samples before the extraction.

The extraction procedure applies only to not-digested samples from which polyphenolic compounds were extracted. The extracts obtained were filtered, dried by evaporation, dissolved in methanol and subjected to further analyses.  In contrast, pasta samples in their natural, solid form were subjected to the digestion process. The protocol used takes into account a constant ratio of meal to digestive fluids, so there was no need to concentrate the solutions. This procedure ensured that the concentration and pH of the final samples were known.

The extraction procedure only applies to samples before digestion, from which polyphenolic compounds were extracted. The extracts were subjected to further analyses. In contrast, pasta samples in their natural, solid form were subjected to the digestion process. The protocol used takes into account a constant ratio of meal to digestive fluids, so there was no need to concentrate the solutions.

-        Material and methods, paragraph 3.8. In vitro Two-Stages Digestion Model: The authors reported an in vitro-two stages protocol, which include gastric and duodenal stage. Moreover, they didn’t perform any investigation about the oral phase, which include the use of “salivary amylase” needed only to digest starch-containing foods. The authors reported a great starch content in the samples (45%, line 87) and the possible linkage of polyphenols to the starch during the digestion (line 293). This experimental approach represents a strong limitation of the entire study. The lack of a preliminary step in the digestion step with a specific action on starch represents a problem in the considering of the effects that this could have in subsequent steps (gastrical and duodenal phases) on the real release of phenolic compounds from the samples. It is therefore strongly recommended repeating of digestion experiments including the oral phase and comparing the results with those reported in the current version of the manuscript.

The authors are very grateful for your valuable guidance, which we will certainly follow in continuing our research. As we were interested from the outset in approximating the bioavailability/released of active compounds, i.e. polyphenols (the degree to which a compounds can be released and absorbed from the gastrointestinal tract ), rather than determining the digestibility of nutrients (the degree to which a nutrient can be broken down into its constituent parts), we chose a simplified protocol in relation to INFOGEST.

We also decided to dispense with the oral phase of digestion because, after initial trials, it became apparent that virtually no changes was taking place in the oral cavity for the polyphenolic compounds we tested. Due to the usually very short interaction of oral enzymes with the food bolus prior to reaching the stomach, their influence is much less clear and rather limited to carbohydrate-rich. For example, it is estimated that only 5% of the consumed starch is  degraded in the mouth cavity by salivary amylase foods  (Hur and others 2011). Therefore A static in vitro digestion model comprising two-stages of digestion (gastric and duodenal) as proposed by the United States Pharmacopeia (2000), and Egger et al. (2016), with some modifications  was applied. We would also like to point out that this type of digestion model has been used by many authors in the context of the study of polyphenolic compounds.

As  mentioned, the method used takes into account some simplifications. However, the sample was initially subjected to a pH of about 2, hence it can be assumed that such conditions may to some extent cause acid hydrolysis of the starch. Under in vivo conditions in the stomach, hydrochloric acid hydrolyzes some carbohydrates to some extent on its own (Kompendium wiedzy o żywnoÅ›ci, żywieniu i zdrowiu, Teresa Mossor-Pietraszewska, Jan GawÄ™cki, PWN 2007). However, at the stage of duodenal digestion, the food sample was treated with pancreatin, which also contains amylase. The main stage of starch digestion occurs in the duodenum – under the influence of pancreatic amylase, contained in pancreatic juice, and intestinal amylase, secreted by small intestinal mucosal cells. They continue the digestive process initiated by salivary amylase. In addition, it is now known that more than 10% of starch is not amylolysed and reaches the colon unchanged and is therefore called resistant starch (Bromatologia. Zarys nauki o żywnoÅ›ci i żywieniu. Henryk Gertig, Juliusz PrzysÅ‚awski. PZWL 2015).

Reviewer 2 Report

Comments and Suggestions for Authors

The Authors present a relevant research topic. Below, they can find several suggestions, for their work

Sentence in L20 seems unfinished. Additionally, the Abstract itself will benefit from a revision

Keywords in their current state repeat the title and are not useful to increasing search results

The Introduction section should be re-arranged as not it lacks consistency. There is no sequence of paragraphs, the aim is not clearly stated at the end of the Introduction

Sentence in L78 is irrelevant. Information about the % of RDA can be more beneficial.

We see no information about the number of repetitions in the Table leading to the SD stated

The paragraph starting in L97 repeats the Introduction, it is unnecessary

How were statistical differences in Tables 2 and 3 obtained, it is not clear at the moment

There is too much information about digestion that is not relevant to the discussion section

The whole discussion lacks consistency. The aim is not fulfilled at all

We see some missing points in the MM explanations related to mainly missing units

The authors state that their research is a pilot design in the conclusion, this should be emphasized right at the beginning of their work

Some issues with formatting references exist.

In MM we see only one sample of pasta in the fiure

Comments on the Quality of English Language

Please refer to sequence to tenses, lexical and typo mistakes.

Author Response

The authors would like to thank the Reviewer for valuable comments which have helped to improve the quality of the manuscript. We hope that the revisions in the manuscript and accompanying responses will be sufficient to make our manuscript suitable for publication. We have made all the changes suggested in the Reviewer's comments in the text.

The Authors present a relevant research topic. Below, they can find several suggestions, for their work. Sentence in L20 seems unfinished. Additionally, the Abstract itself will benefit from a revision

Thank you for your suggestion. The Abstract has been corrected.

Keywords in their current state repeat the title and are not useful to increasing search results

I would like to thank you for your comment. I can confirm that the keywords have been corrected.

The Introduction section should be re-arranged as not it lacks consistency. There is no sequence of paragraphs, the aim is not clearly stated at the end of the Introduction

Thank you for your suggestion. The Introduction has been corrected.

Sentence in L78 is irrelevant. Information about the % of RDA can be more beneficial.

Thank you for your comment. Sentence in L78 has been removed. SD has been converted to RSD%.

We see no information about the number of repetitions in the Table leading to the SD stated

In chapter 3.9. Statistical Analysis provides information on the number of repetitions in the tests conducted during the experiment. Such information is also provided in the caption of the Tables, giving n=3. Information added in the caption to the table, adding the "number of repetitions n=3".

The paragraph starting in L97 repeats the Introduction, it is unnecessary

Thank you for your suggestion. This part of the text has been removed.

How were statistical differences in Tables 2 and 3 obtained, it is not clear at the moment

Thank you for your suggestion. Statistical analysis with ANOVA was applied to determine the significance of differences at α = 0.05; multi-factor analysis of variance and the Tukey test were carried out. The Tukey test is one of the most commonly used tests for comparing pairs of means. It can be used for different sample sizes. It is based on a distribution called the "studentized range statistic." The Tukey method is more conservative than the NIR test, but less conservative than the Scheffé test. The experimental error rate for all pairwise comparisons remains at the set error level, which means that if an α=0.05 level of statistical significance is assumed for the ANOVA test, the same level of statistical significance will be used for all comparisons between pairs (samples).

Information has been added to the captions under the tables about the significance level used in the Tukey test.

There is too much information about digestion that is not relevant to the discussion section. The whole discussion lacks consistency. The aim is not fulfilled at all

Thank you for your suggestion. The discussion has been revised in order to enhance its coherence and to eliminate any superfluous information.

We see some missing points in the MM explanations related to mainly missing units

Thank you for your suggestion. The section Materials and Methods has been corrected.

The authors state that their research is a pilot design in the conclusion, this should be emphasized right at the beginning of their work

Thank you for your suggestion. This information is included at the end of the Introduction.

Some issues with formatting references exist.

Thank you for your suggestion. The References has been corrected.

In MM we see only one sample of pasta in the fiure

Thank you for comment. Another photo was added to the Materials and Methods section.

Round 2

Reviewer 1 Report

Comments and Suggestions for Authors

Drozd et al reported in the manuscript the preparation and the chemical analysis of Acorn flour and pasta enriched with Acorn flour, with a great attention for polyphenols content. This manuscript provides novel insights into enriched pasta preparation and analysis for nutraceutical purposes. However, the manuscript should be revised, as it seems to have unclear experimental details that make the comprehension of the experimental protocols. In conclusion, I believe that this manuscript could give a valid contribution to the scientific field of reference, but a minor revision of the text is required for the manuscript to be accepted. Following I reported my revisions:

1)        Results and discussion, line 121: The authors have supported extensively in their response to my previous reviews about the use of a two-stages gastrosimulated digestion protocol versus the use of the INFOGEST protocol. However, in my opinion, it is necessary to elaborate more about the selection of the two-stages gastrosimulated digestion protocol in the manuscript, in order to clarify the reasons to the reader. The authors may use the data provided in the previous response and the literature references to corroborate the reasons.

2)        Material and methods, paragraph 3.4: The authors should add some details about the HPLC method (e.g., the gradient elution method, the flow rate, and the limits of detection (LODs) and quantifications (LOQs) of phenolic acids).

3)        Material and methods, paragraphs 3.5 and 3.6: Although the authors have reported the bibliographical references of the assays in the manuscript, it would be better to explain in detail the procedure performed, with special attention on incubation times, volumes and concentrations of the reagents used.

4)        Material and methods, paragraph 3.5: Please, indicate the range of concentration, the dilution factors, the LOD, and the LOQ of the gallic acid calibration line.

Comments on the Quality of English Language

A minor editing of English language is required.

Author Response

The authors would like to thank the Reviewer for their constructive comments, which have helped to improve the quality of the manuscript. All the changes suggested in the reviewer's comments have been incorporated into the text.

Drozd et al reported in the manuscript the preparation and the chemical analysis of Acorn flour and pasta enriched with Acorn flour, with a great attention for polyphenols content. This manuscript provides novel insights into enriched pasta preparation and analysis for nutraceutical purposes. However, the manuscript should be revised, as it seems to have unclear experimental details that make the comprehension of the experimental protocols. In conclusion, I believe that this manuscript could give a valid contribution to the scientific field of reference, but a minor revision of the text is required for the manuscript to be accepted. Following I reported my revisions:

1) Results and discussion, line 121: The authors have supported extensively in their response to my previous reviews about the use of a two-stages gastrosimulated digestion protocol versus the use of the INFOGEST protocol. However, in my opinion, it is necessary to elaborate more about the selection of the two-stages gastrosimulated digestion protocol in the manuscript, in order to clarify the reasons to the reader. The authors may use the data provided in the previous response and the literature references to corroborate the reasons.

We would like to thank Reviewer for comment. As suggested by the reviewer, we have discussed in the text the reasons for choosing a two-step digestion protocol instead of a more extensive procedure.

2)        Material and methods, paragraph 3.4: The authors should add some details about the HPLC method (e.g., the gradient elution method, the flow rate, and the limits of detection (LODs) and quantifications (LOQs) of phenolic acids).

3)        Material and methods, paragraphs 3.5 and 3.6: Although the authors have reported the bibliographical references of the assays in the manuscript, it would be better to explain in detail the procedure performed, with special attention on incubation times, volumes and concentrations of the reagents used.

4)        Material and methods, paragraph 3.5: Please, indicate the range of concentration, the dilution factors, the LOD, and the LOQ of the gallic acid calibration line.

Response to comments 2, 3 and 4:  Thank you for your kind comments. We have completed the methodology with the necessary details (sections 3.4-3.6.).

Reviewer 2 Report

Comments and Suggestions for Authors

Authors have responded to comments

Comments on the Quality of English Language

Check for minor mistakes. Authors have responded to previous comments

Author Response

Thank you very much for your comment. We are pleased that we were able to address all the reviewer's objections after the first review and improve the manuscript accordingly.